# Comparing Blend of Essential Oils Plus 25-Hydroxy-Vit-D3 Versus Monensin Plus Virginiamycin Combination in Finishing Feedlot Cattle: Growth Performance, Dietary Energetics, and Carcass Traits

**DOI:** 10.3390/ani12131715

**Published:** 2022-07-02

**Authors:** Alfredo Estrada-Angulo, Daniel A. Mendoza-Cortez, Jorge L. Ramos-Méndez, Yesica J. Arteaga-Wences, Jesús D. Urías-Estrada, Beatriz I. Castro-Pérez, Francisco G. Ríos-Rincón, Miguel A. Rodríguez-Gaxiola, Alberto Barreras, Richard A. Zinn, Alejandro Plascencia

**Affiliations:** 1Faculty of Veterinary Medicine and Zootechnics, Autonomous University of Sinaloa, Culiacan 80260, Sinaloa, Mexico; alfred_vet@hotmail.com (A.E.-A.); danielmendozac@outlook.com (D.A.M.-C.); ramos.jorge.92@outlook.com (J.L.R.-M.); arteaga.yesi.92@hotmail.com (Y.J.A.-W.); uriasestrada_jd@hotmail.com (J.D.U.-E.); laisa_29@hotmail.com (B.I.C.-P.); fragerr@hotmail.com (F.G.R.-R.); m-angel@uas.edu.mx (M.A.R.-G.); 2Veterinary Science Research Institute, Autonomous University of Baja California, Mexicali 21100, Baja California, Mexico; beto_barreras@yahoo.com; 3Animal Science Department, University of California, Davis, CA 95616, USA; razinn@ucdavis.edu

**Keywords:** feedlot cattle, essential oils, vitamin D3, monensin, virginiamycin, growth performance, carcass

## Abstract

**Simple Summary:**

A primary objective in livestock production is the limitation of use of conventional antibiotics as feed additives for promoting growth. Due to their antimicrobial characteristics, essential oils are “generally recognized as safe” alternatives to conventional antibiotics. In this study, dietary supplementation of finishing cattle with essential oils combined with vitamin D3 improved dietary energy utilization comparable to that of the combination of ionophore monensin and antibiotic virginiamycin. Furthermore, supplementation with essential oils combined with vitamin D3 during the finishing phase may improve carcass Longissimus muscle area and carcass yield.

**Abstract:**

Ninety crossbreed bulls (349.5 ± 8.25 kg initial weight) were used in an 87day trial to compare the effects of a blend of essential oils plus 25-hydroxy-Vit-D3 (EO + HyD) versus the combination of monensin with virginiamycin (MON + VM) on feedlot growth performance and carcass characteristics. Dietary treatments (nine replicates/treatment) were supplemented with 40 mg/kg diet dry matter of MON + VM (equal parts) or with 120.12 mg/kg diet dry matter of a combination of standardized mixture of essential oils (120 mg) plus 0.12 mg of 25-hydroxy-vitamin-D3 (EO + HyD). There were no treatment effects on dry matter intake (DMI, *p* = 0.63). However, the coefficient of variation in day-to-day DMI was greater for EO + HyD than for MON + VM (11.4% vs. 3.88%, *p* = 0.04). There were no treatment effects (*p* ≥ 0.17) on daily weight gain, gain-to-feed ratio, and estimated dietary net energy. Cattle supplemented with EO + HyD had greater Longissimus muscle area (7.9%, *p* < 0.01) and estimated retail yield (1.6%, *p* = 0.03), and tended to have heavier (1.7%, *p* = 0.10) carcass weight. Differences among treatments in dressing percentage, fat thickness, kidney–pelvic–heart fat, and marbling score were not appreciable (*p* > 0.10). It is concluded that growth performance response and dietary energetic are similar for finishing cattle supplemented with EO + HyD vs. MON + VM. However, compared with MON + VM, supplementation with EO + HyD during the finishing phase may improve carcass Longissimus area and carcass yield.

## 1. Introduction

Virginiamycin (VM), a peptolide antibiotic, has been used as feed additive in feedlot cattle to decrease the incidence of acidosis and liver abscess, and to promote growth [1,2,3]. The effect of VM on Gram-positive bacteria is similar to that of ionophore monenesin (MON), although their modes of action differ (virginiamycin inhibits protein synthesis by binding to the 23S ribosomal subunit blocking the translation process, with no effect on transcription, while the ionophore monensin the modifies movement of ions across the membranes of rumen microbes) [4]. Accordingly, numerous studies [5,6,7,8] have demonstrated an additive effect when the two antibiotics are fed in combination. Concern over routine use of supplemental antibiotics for growth enhancement has led to the search nonantibiotic alternatives that may likewise enhance growth performance [9]. Supplemental essential oils (EOs; such as piperine, limonene, eugenol, and thymol) also manifest antimicrobial properties by modifying ruminal fermentation patterns (greater molar ratio of ruminal propionate to acetate), retarding the rate of ruminal starch fermentation, and reducing ruminal degradation of feed protein [10,11]. In this regard, supplementation of feedlot finishing diets with a standardized mixture of essential oils compounds (thymol, eugenol, vanillin, and limonene) at dosages of 90 to 150 mg/kg diet dry matter (DM) resulted in enhancements in gain efficiency and efficiency of energy utilization comparable to those of MON or VM supplementation separately [12,13]. In recent reports, 25-hydroxy-vitamin-D3 (HyD) supplemented at 0.10 mg/kg of diet tended to increase average daily gain (ADG) [14], carcass weight [15], and dressing percentage [16]. Researchers attributed these results to the effects of HyD on protein accretion [17]. Due to the chemical nature and the mechanism of action of EO and HyD, their combination could be complementary. We hypothesized that supplementation of feedlot cattle finishing diets with the combination of EO plus HyD may result in enhancements in growth performance and energetic efficiency comparable to the combination of MON with VM.

## 2. Materials and Methods

The trial was conducted at the feedlot research facilities located in Guasave, Sinaloa, México (25°33′ N and 108°25′ W). The site is about 50 m above sea level and has a dry climate. During the course of the experiment, ambient air temperature averaged 22.6 °C (minimum and maximum of 16.3 and 28.0 °C, respectively), and relative humidity averaged 53.9% (minimum and maximum of 18.4% and 88.0%, respectively). All animal management procedures were conducted within the guidelines of locally approved techniques for animal use and care [18]. The experimental protocol was approved by the Universidad Autonoma de Sinaloa Animal Use and Care Committee (Protocol #21092021).

### 2.1. Animal Processing, Diets, Treatments, Housing, and Feeding

Ninety intact male (349.5 ± 8.25 kg initial shrunk weight) crossbreed cattle (approximately 50% Zebu breeding with the remainder represented by continental and British breeds in various proportions) were used to evaluate the treatments effects on growth performance, dietary energetic, and carcass characteristics. On arrival into the feedlot (approximately 3 months before initiation of the experiment), cattle were vaccinated for bovine rhinotracheitis and parainfluenza 3 (TSV-27, Pfizer Animal Health, México City, Mexico), clostridials (Fortress 7, Pfizer Animal Health, Mexico City, Mexico), and *Pasteurella haemolytica* (One Shot, Pfizer Animal Health, México City, Mexico), and treated against parasites (CYDECTIN^®^ NF, Pfizer Animal Health, México; Trodax, Merial, México). Cattle were injected with 500,000 IU of vitamin A, 75,000 IU of vitamin D3, and 50 IU of vitamin E (Synt-ADE, Zoetis) and were implanted with 40 mg of trenbolone acetate and 8 mg of estradiol 17β (Revalor G, MSD Salud Animal Mexico, Santiago Tianguistenco, México). Cattle were fed transition diets for 70 days prior to receiving the finishing diet. Cattle were adapted to the basal finishing diet (minus treatment additives) for 2 weeks prior to initiation of the study. The ingredient composition of the basal diet and its chemical composition according to NASEM [19] are shown in Table 1. Upon initiation of the study, cattle were implanted with 140 mg of trenbolone acetate and 20 mg of estradiol 17β (Revalor, MSD Salud Animal Mexico, Santiago Tianguistenco, México) and individually weighed (feed and water were not withdrawn before weighing). Cattle were blocked by weight and randomly allocated within blocks to two treatments (nine pens/treatment, five bulls/pen). Pens were 5.00 × 12.00 m with 19 m^2^ of shade and were equipped with automatic waterers and fence-line feed bunks (2.37 m in length). Experiments lasted 87 days. Dietary treatments consisted of a steam-flaked corn-based finishing diet (Table 1) supplemented (dry matter basis, DM) with (1) MON + VM, consisting of 20 mg monensin/kg diet (MON; Rumensin, Elanco Animal Health, Indianapolis, IN, USA) plus 20 mg virginiamycin/kg diet (VM; V-max 50, Phibro Animal Health, Ridgefield Park, NJ, USA) or (2) EO + HyD, consisting of a 120 mg of a blend of essential oils/kg diet (EO; CRINA, DSM Nutritional Products, Basel, Switzerland) plus 0.12 mg of 25-hydroxy-vitamin-D3/kg diet DM (Hy-D; DSM Nutritional Products, Basel, Switzerland). Diets were prepared at weekly intervals. Daily feed allotments to each pen were adjusted to allow minimal (<5%) feed refusals. Amounts of feed offered and of feed refused were weighed daily. Cattle were provided fresh feed twice daily at 8:00 a.m. and 2:00 p.m. in a 40:60 proportion (as fed basis). Feed bunks were visually assessed between 7:00 and 7:30 a.m. each morning, refusals were collected and weighed, and feed intake was determined. Adjustments to daily feed delivery were provided at the afternoon feeding. Cattle were fasted (drinking water was not withdrawn) for approximately 16 h before recording the final live weight (LW).

### 2.2. Laboratory Analyses

Feed and refusal samples were collected daily for dry matter analysis (oven drying at 105 °C until no further weight loss occurred; method 930.15 [20].

### 2.3. Calculations

Estimations of expected dry matter intake (DMI) and dietary net energy value were performed on the basis of measures of initial shrunk body weight (SBW), assuming that SBW is 96% of full weight [21], and fasted (16 h) final weight. Average daily gain (ADG) was computed by subtracting the initial SBW from the final fasted weight and dividing the result by the number of days on feed (87 days). Gain efficiency was computed by dividing ADG by the daily DMI. The estimation of expected DMI was performed on the basis of the observed ADG and SBW according to the following equation: expected DMI, kg/day = (EM/NE_m_) + (EG/NE_g_), where EM (energy required for maintenance, Mcal/day) = 0.077W^0.75^, EG (energy required for gain, Mcal/day) = ADG^1.097^ × 0.0557W^0.75^ [22], and divisors NE_m_ (diet energy for maintenance) and gain NE_g_ (diet energy for gain) = 2.14 and 1.46 Mcal/kg (derived from tabular values based on the ingredient composition of the experimental diet [19]). Estimation of dietary NE_m_ was performed by means of the following quadratic formula:$$x=\frac{-b\pm \sqrt{b^{2}-4ac}}{2c},$$
where *x* = NE_m_, Mcal/kg, *a* = −0.41EM, *b* = 0.877EM + 0.41DMI + EG, and *c* = −0.877 DMI [23]. The observed dietary NE_g_ was derived from observed dietary NE_m_ using the equation NE_g_ = 0.877NE_m_ − 0.41 [24].

### 2.4. Carcass Evaluation

All steers were harvested on the same day at a government-certified (TIF) commercial abattoir. Hot carcass weights (HCWs) were obtained from all steers at the time of harvest. After carcasses were chilled at −2 °C to 1 °C for 48 h, the following measurements were obtained: (1) Longissimus muscle area (LM), taken by direct grid reading at the 12th rib; (2) subcutaneous fat over the LM muscle at the 12th rib taken at a location three-quarters of the lateral length from the chin bone end; (3) kidney, pelvic, and heart fat (KPH) as a percentage of carcass weight; (4) marbling score [25]. Estimated retail yield of boneless, closely trimmed retail cuts from the round, loin, rib, and chuck (% of HCW) was estimated as follows: retail yield = 52.56 − 1.95 × subcutaneous fat − 1.06 × KPH + 0.106 × LM area − 0.018 × HCW [25].

### 2.5. Statistical Analyses

Growth performance data (gain, gain efficiency, and dietary energetics) were analyzed as a randomized complete block design, with pen as the experimental unit. Carcass data were analyzed using the MIXED procedure [26], with treatment and pen as fixed effects and interaction treatment × pen and individual carcasses within pen by treatment subclasses as random effects. For comparing DM intake pattern, equality of mean effects and homogeneity between variances (CV1 vs. CV2) were tested using Brown and Forsythe’s variation of Levene’s test. In all cases, the least squares mean and standard error are reported; contrasts were considered significant when *p* ≤ 0.05, and tendencies were identified when 0.05 < *p* ≤ 0.10.

## 3. Results

There was no morbidity or mortality during the course of the study. On the basis of measures of feed intake (Table 2), average daily intake of the combination of MON + VM was 326 mg/day (equivalent to 0.787 mg/kg LW). Average daily intake of EO was 948 mg/day (2.3 mg/kg LW), and intake of HyD averaged 0.95 mg/day (0.0023 mg/kg LW).

Treatment effects on growth performance and dietary energetics are shown in Table 2. There was no treatment effect on dry matter intake (DMI, *p* = 0.64) averaging 7.91 ± 0.617 kg/day. However, the coefficient variation in day-to-day DMI was greater (294%, *p* < 0.04) for EO + HyD than for MON + VM. There were no treatment effects on daily gain (*p* = 0.67), gain-to-feed ratio (*p* = 0.17), and estimated dietary net energy values (*p* = 0.86). Estimated dietary net energy based on growth performance averaged 5.4% greater than expected according to tabular values [19] and diet formulation (Table 1).

Treatment effects on carcass characteristics are shown in Table 3. Cattle supplemented with EO + HyD had greater LM area (7.9%, *p* < 0.01) and estimated retail yield (1.6%, *p* = 0.03), and they tended to have heavier (1.7%, *p* = 0.10) carcass weight. Differences among treatments in dressing percentage, fat thickness, KPH, and marbling score were not significant (*p* > 0.10).

## 4. Discussion

The daily dose of MON plus VM (0.787 mg/kg LW) was within the suggested dosage range for increased average daily gain and gain-to-feed ratio when both additives are fed in combination [8,27]. Optimal dosage of EO + HyD has not been established for feedlot cattle. However, the daily dosages of EO (3 mg/kg LW) and HyD (0.003 mg/kg LW) were within the range of levels previously shown to enhance growth performance and/or carcass characteristics when offered separately in high-energy finishing diets [15,28].

One of the reasons for ionophore supplementation of high-energy finishing diets is a reduction in the variation of daily feed intake [29,30], as day-to-day variation in feed intake is thought to be a contributing factor to subclinical acidosis [31,32]. Consistent with the present study, Barreras et al. [33] observed a threefold reduction in DMI variation for cattle fed MON vs. non-supplemented diets. Variations in DMI of up to 15% did not appreciably affect ADG and gain efficiency [34]. In the same manner, Barajas et al. [35] observed that day-to-day DMI fluctuations up to 20% did not affect gain efficiency. Accordingly, it may be expected that the intake variation of 11.4% observed for the EO + HyD combination was not sufficient to appreciably affect growth performance.

When supplemented alone, MON decreases DM intake by an average of 3.1% [36], whereas VM supplementation alone does not reduce DMI [37]. There are few reports evaluating the effects of MON + VM combination on cattle growth performance. However, consistently with the present study, most reports [5,6,38] found that the combination did not affect daily DMI, although Benatti et al. [27] observed decreased DMI in feedlot cattle supplemented with MON + VM.

As mentioned previously, there are no previously reported studies evaluating the effects of EO + HyD combination on feedlot cattle growth performance. Essential oils are, by nature, volatile and aromatic and, as such, are claimed to enhance diet acceptability, promoting voluntary feed intake [39]. However, when supplemented separately, the blend of EO used in the present study did not increase the feed intake of feedlot cattle [40,41,42] or feedlot lambs [13,43]. Similarly, HyD supplementation alone did not increase DMI of feedlot cattle [15,16].

Except for Rigueiro et al. [37], the majority of studies reported enhancements in daily gain and gain efficiency for the combination of MON + VM than when supplemented separately [3]. The positive effects of MON and VM on energetic efficiency in ruminants have been extensively studied. The main effects include reductions in acetate-to-propionate ratio and methane production, decreased ruminal protein degradation, increased N retention, and improved intestinal health, promoting greater nutrient absorption [44,45,46]. Essential oils, likewise, have antimicrobial properties comparable to those of ionophores and VM. Additionally, they are observed to enhance immune response and reduce cellular oxidative stress [11,47,48]. Vitamin D3 at high levels of supplementation (i.e., ≥1 mg/animal/day) is active in immune and endocrine functions [49] and in mechanisms that promote protein accretion [17]. All of these enhancements associated with supplemental MON, VM, EO, and HyD help to facilitate feedlot cattle energetic efficiency.

Estimation of dietary net energy based on measures of growth performance (observed dietary NE) permits the comparison with expected based on diet composition and tabular feed standards [19]. A ratio of observed-to-expected dietary NE ratio of 1.00 indicates that growth performance (daily gain) is consistent with theoretical dietary NE values (based on feed standard tables [19], Table 1) and observed DMI. A ratio greater than 1.00 is indicative of greater dietary energy utilization efficiency, while an observed-to-expected ratio less than 1.00 indicates less efficient dietary energy utilization. This study did not include a negative control (with no treatment additives); thus, direct net additive responses are uncertain. However, on the basis of observed-to-expected dietary net energy, efficiency of energy utilization for cattle fed MON + VM was 5.6% greater than expected, while energetic efficiency of cattle receiving the EO + HyD combination was 5.2% greater than expected. The observed increase in the ratio of observed-to-expected dietary net energy for MON + VM combination is in close agreement with the increase of 5.9% reported by Nuñez et al. [50] for the combination of VM plus the ionophore salinomycin. According to meta-analyses, Duffield et al. [36] observed that supplemental MON alone increase efficiency by an average of 3.5%. The average increase in energetic efficiency due to supplemental VM alone averaged 4.1% [1,3,45,51,52]. Therefore, the increase in efficiency above the expected values when both additives are supplemented separately could confirm the complementary effect of the combination observed both in this experiment and in previous reports [6,27,53]. On the other hand, using performance data from several reports in feedlot cattle [12,14,40,41,54,55], the average increase of estimated dietary net energy due to supplemental EO alone was 3.6%. Supplemental HyD, alone, has not been found to affect efficiency of energy utilization [15,56].

When supplemented alone, the effects of EO on carcass characteristics was not appreciable in feedlot cattle [12,57] and finishing lambs [13,43]. The NASEM [19] does not report vitamin D equivalents for a majority of dietary ingredients. Consequently, it may be considered that supplemental HyD represents the total quantity of vitamin D ingested. Supplementation with HyD at 0.12 mg vitamin D/kg diet DM (equivalent to 4800 IU vitamin D3/kg diet DM) is equivalent to 2.2-fold the suggested allowance [58]. High-level short-term (i.e., 10 days) supplementation with vitamin D3 prior to slaughter has been suggested as a means for increasing calcium uptake, thereby enhancing meat tenderness [59]. However, effects on meat tenderness were not appreciable [60] when D3 was supplemented at high levels for longer durations (i.e., 67 days). More recently [49,61], the importance of additional vitamin D3 supplementation for livestock health and productivity was highlighted. However, the benefits of additional vitamin D3 supplementation to feedlot cattle on growth performance and carcass characteristics are inconsistent. Dosages of up to 125 mg/animal/day for 8 days previous to slaughter had negative effects on DMI, ADG, and feed efficiency [62]. However, daily dosage of 12 mg/day or lower did not adversely affect performance or carcass characteristics [62,63]. Long-term vitamin D3 supplementation (160 days) at moderate levels (from 0.04 to 0.05 mg/day) was not beneficial for enhancement of cattle growth performance or carcass characteristics [64,65]. In contrast, daily supplementation at higher levels (~1 mg/animal) increased carcass dressing percentage (+0.54 percentage points) [16] and carcass weight (+1.49%) [15] in Nellore cattle during a finishing period of at least 90 days. These effects can be partially explained by the potentiating effects of higher-level vitamin D3 supplementation on protein accretion [17]. The basis for the increase in LM area in cattle fed EO + HyD is not clear. In earlier studies, supplemental EO alone [28,66] or supplemental HyD alone did not affected LM area [14,15,16]. More research related to this topic is necessary to corroborate the results obtained herein.

## 5. Conclusions

It is concluded that growth performance responses and dietary energetics are similar for finishing cattle supplemented with EO + HyD vs. MON + VM. However, compared with MON + VM, supplementation with EO + HyD during the finishing phase may improve carcass LM area and carcass yield.

## Figures and Tables

**Table 1 animals-12-01715-t001:** Composition of basal diet fed by cattle.

	Treatments	
Item	MON + VM	EO + HYD
Ingredient composition (% DM basis)		
Corn stover	12.00	12.00
Steam-flaked corn	67.80	67.80
Soybean meal (Ref 5-20-637) [19]	6.00	6.00
Premix dilution MON + VM ^§^	1.50	----
Premix dilution EO + HyD ^†^	---	1.50
Molasses cane	7.50	7.50
Yellow grease	2.70	2.70
Agromix SP ^‡^	2.50	2.50
Chemical composition (%DM basis) ^⁋^		
Dry matter	84.60	84.60
Crude protein	11.80	11.80
Rumen degradable protein (as % of protein)	59.82	59.82
Neutral detergent fiber	15.36	15.36
Calcium	0.81	0.81
Phosphorus	0.30	0.30
Calculated net energy (Mcal/kg)		
Metabolizable energy	3.05	3.05
Maintenance	2.14	2.14
Gain	1.46	1.46

^†^ Agromix SP contained the following: CP, 53.0%; calcium, 13.6%; phosphorus, 0.40%; magnesium, 1.0%; potassium, 0.71%; NaCl, 15%; Co, 5.59 ppm; Fe, 2759 ppm; Zn, 2913 ppm; Cu, 20 ppm; Mn, 1674 ppm; vitamin A, 225 IU/g; vitamin E, 1.26 UI/g (Agronutrientes del Norte, Monterrey, NL, Mexico). ^§^ Premix dilution contained the following (per 10 kg): calcium, 20.9%; phosphorus, 3.9%; Vit A, 40,000 KUI; Vit D3, 50,000 KUI; biotin, 1.87 ppm; Lactonúcleo industrial (trace mineral), 50 g/kg; 13 g of monensin/kg diet plus 13 g of virginiamycin/kg diet. ^‡^ Premix dilution contained the following (per 10 kg): calcium, 20.9%; phosphorus, 3.9%; Vit A, 40,000 KUI; Vit D3, 50,000 KUI; biotin, 1.87 ppm; Lactonúcleo industrial (trace mineral), 50 g/kg; 67 g of EO plus 0.54 g of HyD/kg diet (CRINA Ruminants and Hy-D; DSM Nutritional Products, Basel, Switzerland, Nutritional Products, Basel, Switzerland). ^⁋^ Nutrient composition (except DM which was determined in laboratory) and net energy values are based on the diet formulation and tabular values for individual feed ingredients [19].

**Table 2 animals-12-01715-t002:** Effect of treatments on growth-performance and dietary energy of finishing cattle.

	Treatments ^†^			
Item	MON + VM	EO + HyD	SEM	*p*-Value
Days on test	87	87		
Pen replicates	9	9		
Live weight, kg/day ^§^				
Initial	349.43	349.74	3.153	0.98
Final	472.80	474.41	5.180	0.69
Average daily gain, kg/day	1.418	1.433	0.041	0.67
Dry matter intake, kg/day	7.855	7.965	0.202	0.64
Daily DM intake variation,%	3.88	11.37	3.96	0.04
Feed efficiency, kg/kg	0.181	0.180	0.002	0.17
Diet net energy, Mcal/kg				
Maintenance	2.26	2.25	0.023	0.67
Gain	1.57	1.56	0.017	0.67
Observed-to-expected diet NE, Mcal/kg				
Maintenance	1.056	1.052	0.009	0.67
Gain	1.076	1.070	0.012	0.67
Observed-to-expected DMI	0.936	0.940	0.009	0.67

^†^ MON = monensin 20 g/ton; MON + VM = combined 20 g monensin + 20 g virginiamycin/ton; EO + HyD = standardized source of a mixture of essential oils (CRINA Ruminants; DSM Nutritional Products, Basel, Switzerland) plus 25-hydroxy-vitamin-D3 (Hy-D; DSM Nutritional Products, Basel, Switzerland) formulated to provide 120 mg/kg diet EOC plus 0.12 mg/kg diet HyD. ^§^ Initial shrunk weight is the full live weight reduced 4% to adjustment for gastrointestinal fill; final weight was registered after 16 h of fasting.

**Table 3 animals-12-01715-t003:** Effect of treatments on carcass characteristics of finishing cattle.

	Treatments ^†^			
Item	MON + VM	EO + HyD	SEM	*p*-Value
Hot carcass weight, kg	300.93	306.13	2.89	0.10
Dressing percentage	63.65	64.53	0.47	0.17
Cold carcass weight, kg	297.87	301.53	2.88	0.12
Cooler shrink, %	1.38	1.51	0.091	0.32
LM area, cm^2^	82.77	89.89	2.39	<0.01
Fat thickness, cm	0.74	0.69	0.042	0.39
Kidney pelvic and heart fat, %	1.73	1.65	0.050	0.53
Marbling score ^§^	171	188	12.1	0.33
Retail yield *	52.63	53.51	0.21	0.03

^†^ MON = monensin 20 g/ton; MON + VM = combined 20 g monensin + 20 g virginiamycin/ton; EO + HyD = standardized source of a mixture of essential oils (CRINA Ruminants; DSM Nutritional Products, Basel, Switzerland) plus 25-hydroxy-vitamin-D3 (Hy-D; DSM Nutritional Products, Basel, Switzerland) formulated to provide 120 mg/kg diet EOC plus 0.12 mg/kg diet HyD. ^§^ Estimated marbling grade coded as standard <300, minimum slight = 300, or minimum small = 400. * Estimated retail yield of boneless, closely trimmed retail cuts from the round, loin, rib, and chuck (% of HCW) was estimated as follows: retail yield = 52.56 − 1.95 × subcutaneous fat − 1.06 × KPH + 0.106 × LM area − 0.018 × HCW [25].

## Data Availability

The data presented in this study are available on request from the corresponding author.

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
