# Peer review of "Comparing Blend of Essential Oils Plus 25-Hydroxy-Vit-D3 Versus Monensin Plus Virginiamycin Combination in Finishing Feedlot Cattle: Growth Performance, Dietary Energetics, and Carcass Traits"

_animals, 2022, doi:10.3390/ani12131715_

Round 1

Reviewer 1 Report

1)In the introduction and discussion line 262 it is stated that a combination of EO and HyD could be synergistic. Synergistic effects are when the sum of the effects is more than the two individual effects combined,i.e 2+2=10. Additive effects are when the sum of the effects equals the the two individual effects combined, i.e 2+2=4. Complimentary effects are when the sum of effects is more than any of the individual effects, i.e. 2+2=3. It might be more appropriate to refer to complimentary effects due to different modes of action.

2)Expt design and treatments: A control diet supplemented with monensin only would have strengthened the scientific value of the study and would have enabled the authors to calculate a benefit to cost ratio which is important to nutritionists and industry in general. On the other hand facilities and availability of animals are constraints for many research institutes and university research farms.

3)Table 1:

a) Soybean meal should be described in more detail. In the NASEM (2016) composition of feeds tables there are three different soybean meals listed namely SBM low CP (46% CP, 55% RDP),  SBM high CP(53% CP, 70% RDP) and SBM heated (48% CP, 50% RDP). The type of soybean meal can therefore have an effect on dietary CP and RDP. The addition of RDP to the table can be considered as it is an indication of sufficient substrates for microbial prot production. Some countries use the ME system and those readers are more familiar with ME values, it can also be added to the chemical composition.

b)If available authors should consider performing wet chemical analyses on the feed samples and orts since it can differ significantly from the feed tables.The orts can be analysed for NDF and CP only as it can give an indication of selective feed consumption.

c)The dietary Ca content is 0.81% which is double the NASEM recommendation for cattle  weighing 400kg and gaining 1.41kg/d Is there any explanation for this?

3)Line 175: DMI P=0.64 not 0.63 and Line 177 P=0.67 not 0.49

4)Line 274: It is mentioned that short term supplementation with vit D3 prior to slaughtering can enhance meat tederness. Why was the Warner Braztler sheare force not measured.?

5)The role of vitD3 is not only limited to Ca homeostasis and bone metabolism but is also associated with immunity. The number of animals pulled for treatment as well as the incidence of metabolic disorders can be reported if available.

Author Response

We are grateful to reviewers for the time and effort in helping improve the quality of the manuscript. The observations were wise and timely which permit the improvement substantially the manuscript. We have addressed the concerns in our revised manuscript accordingly.

All changes and correction made are highlighted in yellow in the corrected version of the manuscript.

Reviewer 1

RW: In the introduction and discussion line 262 it is stated that a combination of EO and HyD could be synergistic. Synergistic effects are when the sum of the effects is more than the two individual effects combined, i.e 2+2=10. Additive effects are when the sum of the effects equals the two individual effects combined, i.e 2+2=4. Complimentary effects are when the sum of effects is more than any of the individual effects, i.e. 2+2=3. It might be more appropriate to refer to complimentary effects due to different modes of action.

AU: Totally agree. The term “synergistic” was changed as suggested.

RW: Experiment design and treatments: A control diet supplemented with monensin only would have strengthened the scientific value of the study and would have enabled the authors to calculate a benefit to cost ratio which is important to nutritionists and industry in general. On the other hand, facilities and availability of animals are constraints for many research institutes and university research farms.

AU: We agree. We performed previous studies comparing monensin alone vs blended oils. These comparisons were conducted using lambs and feedlot cattle (one published in small ruminant research) and the other under review. This background is mentioned in the Introduction section. Due to facility constraints, we decided go with the two treatments, allowing sufficient power for pertinent inferences.

RW: Table 1:

  1. Soybean meal should be described in more detail. In the NASEM (2016) composition of feeds tables there are three different soybean meals listed namely SBM low CP (46% CP, 55% RDP), SBM high CP (53% CP, 70% RDP) and SBM heated (48% CP, 50% RDP). The type of soybean meal can therefore have an effect on dietary CP and RDP. The addition of RDP to the table can be considered as it is an indication of sufficient substrates for microbial prot production. Some countries use the ME system and those readers are more familiar with ME values, it can also be added to the chemical composition.

AU: All suggestions were covered in Table 1.

  1. If available authors should consider performing wet chemical analyses on the feed samples and orts since it can differ significantly from the feed tables. The orts can be analysed for NDF and CP only as it can give an indication of selective feed consumption.

AU: Analysis of orts are performed in digestion trials where animals were fed ad libitum, and as you indicate, orts analysis is used as tool to determine selective consumption (ingredient or particle size). Both circumstances were not objectives in our experiment.

  1. The dietary Ca content is 0.81% which is double the NASEM recommendation for cattle weighing 400kg and gaining 1.41kg/d Is there any explanation for this?

AU: There is considerable variation in the application of mineral “requirements” by feedlot nutritionists. In a feedlot survey (Vasconcelos and Galyean, 2007) the Ca concentration of commercial feedlot diet formulations ranged from 0.6 to 0.9% Ca (DM basis). Although studies have shown enhanced ADG with increasing Ca levels above NASEM recommendations (Bock et al., 1991, Buenabad et al 2020), the basis for the perceived benefit may be only partially a consideration of “requirements”. Huntington (1983) observed that the increased ADG with increasing level of limestone supplementation may have had more to do with a buffering effect than Ca requirement, per se. With this in mind, the vast majority of our published research with feedlot cattle involve diets formulated to contain about 1.5% limestone, corresponding to 0.8% total dietary Ca.

RW: Line 175: DMI P=0.64 not 0.63 and Line 177 P=0.67 not 0.49

AU: Thanks!! Correction were made

RW: Line 274: It is mentioned that short term supplementation with vit D3 prior to slaughtering can enhance meat tenderness. Why was the Warner Braztler sheare force not measured.?

AU: Long-term supplementation at high levels (i.e., 67-d; Rodriguez et al., 2013) has not resulted in appreciable changes in meat tenderness. Our experiment was a long-term duration (84-d). Clarification was inserted in the corrected version of the manuscript.

RW: The role of vitD3 is not only limited to Ca homeostasis and bone metabolism but is also associated with immunity. The number of animals pulled for treatment as well as the incidence of metabolic disorders can be reported if available.

AU: Fortunately, we did not have any health complications. Mention of this was included the revision.

Reviewer 2 Report

Well written text, with quotes from other authors that qualify the result found by the research.

The results are presented objectively and the discussion takes place in a didactic way, favoring the understanding and facilitating the reading of the article. A point that can be improved is the search for more current references, as an example we have the reference number four, which could be replaced by a more recent one.

In lines 49-50, in the passage "although their modes of action differ" the similarities and differences between the mechanisms of action could have been mentioned.

Author Response

We are grateful to reviewers for the time and effort in helping improve the quality of the manuscript. The observations were wise and timely which permit the improvement substantially the manuscript. We have addressed the concerns in our revised manuscript accordingly.

All changes and correction made are highlighted in yellow in the corrected version of the manuscript.

Reviewer 2

RW: Well written text, with quotes from other authors that qualify the result found by the research.

AU: Thanks for your kind comment

RW: The results are presented objectively and the discussion takes place in a didactic way, favoring the understanding and facilitating the reading of the article. A point that can be improved is the search for more current references, as an example we have the reference number four, which could be replaced by a more recent one.

AU: Following your suggestion, the reference was replaced by a more recent one.

RW: In lines 49-50, in the passage "although their modes of action differ" the similarities and differences between the mechanisms of action could have been mentioned.

AU: Mechanism of action of both additives are now mentioned in the revision.

Reviewer 3 Report

This is a very interesting study. The design is appropriate. All sections were well described and overall idea about this study is important for the livestock producers and consumers a like. 

Author Response

Reviewer 3

RW: This is a very interesting study. The design is appropriate. All sections were well described and overall idea about this study is important for the livestock producers and consumers alike. 

AU: Thanks for your kind comment!

Reviewer 4 Report

The manuscript entitled "Comparing blend of essential oils plus 25-hydroxy-Vit-D3 versus monensin plus virginiamycin combination in finishing feedlot cattle: Growth performance, dietary energetics, and carcass traits" has been reviewed. Despite ambitious title, it is hard to support its conclusions in the current way for the following reasons:

(1) Experiment design: It is obvious to see that the current study was lack of control treatment, at least the control design for essential oils is needed.

(2) Data size: It is also easy to see from the title (Growth performance, dietary energetics, and carcass traits) that only little data were showed in the current work. It is confirmed from the section of Result that only Table 2: Effect of treatments on growth-performance and dietary energy of finishing cattle and Table 3: Effect of treatments on carcass characteristics of finishing cattle, showed the related results. The data size is really small for a scientific experiment, and I personally think more data should be provided to support the conclusion.

(3) Methods: With ninety intact male were used in this study, and they were fed in pens. It is necessary to clarify the detailed methods for the growth performance and carcass characteristics treatment, n=9 or n=45, or others?

(4) Others: Line 132-134: Estimation of expected dry matter intake (DMI) and dietary net energy value were performed based on measures of initial shrunk body weight (SBW), assuming that SBW is 96% of full weight [21]. I have noticed that reference 21 was NRC (2000), why you did not take NRC (2016) even new version has been released? Many changes have been occurred batween NRC (2000) and NRC (2016). For references section, Line 478-479: Pukrop, J. R.; Campbell, B.T.; Schoonmaker, J.P. Effect of essential oils on performance, liver abscesses, carcass characteristics, 478 and meat quality in feedlot steers. Anim. Feed Sci. Technol. 2019, 257: 11496. https://doi.org/10.1016/j.anifeedsci.2019.114296, why here 2019 was not bold? Such mistakes should be checked throughout the text.

Author Response

Reviwer 4

We are disagree with the majority of your observations. You can find concrete arguments for each reason exposed in the attached file

Round 2

Reviewer 4 Report

I also disgaree with your response. Please see following reasons:

(1) Experiment design: I did not think Monensin plus VM is actually a control, as you described as positive control.

(2) Data size: I mean the amount of data, not the number of replication, was really small.  There were only two tables to show your results, without any figure! In your response, why not list some similar work in a detailed way? More data content should be provided to support your conclusion.

(3) Data accuracy: Table 3, it is hard to believe that the LM area in EO+HyD 89.89 should be higher than that in MON+VM 82.77 with the P value <0.01! For Daily DM intake variation, how comes the sigificance between 3.88 and 11.37 with SEM of 3.96?

(4) Conclusion: Have you compared the cost of these two treatments? What is the significance of current study in field of animal science? Only two indexes, i.e., carcass LM area and carcass yield, showed positive effect, which one did you recommend? These issues should be carefully treated when conducting a applied experiment!